# Particular Aspects Related to CD4+ Level in a Group of HIV-Infected Patients and Associated Acute Coronary Syndrome

**DOI:** 10.3390/diagnostics13162682

**Published:** 2023-08-15

**Authors:** Mircea Bajdechi, Adriana Gurghean, Vlad Bataila, Alexandru Scafa-Udriște, Georgiana-Elena Bajdechi, Roxana Radoi, Anca Cristiana Oprea, Valentin Chioncel, Iuliana Mateescu, Lucia Zekra, Roxana Cernat, Irina Magdalena Dumitru, Sorin Rugina

**Affiliations:** 1Faculty of Medicine, University of Medicine and Pharmacy “Carol Davila” of Bucharest, 050474 Bucharest, Romania; mircea.bajdechi@gmail.com (M.B.); alexscafa@yahoo.com (A.S.-U.); georgianastoian1993@gmail.com (G.-E.B.); crisoprea2512@yahoo.com (A.C.O.); valentin.chioncel@umfcd.ro (V.C.); iuliamateescu07@gmail.com (I.M.); 2Doctoral School of Medicine, “Ovidius” University of Constanta, 900470 Constanta, Romania; luciazekra@yahoo.com (L.Z.); r.cernat@seanet.ro (R.C.); dumitrui@hotmail.com (I.M.D.); sorinrugina@yahoo.com (S.R.); 3Emergency Clinical Hospital of Bucharest, 014461 Bucharest, Romania; 4Clinical Hospital of Infectious and Tropical Disease “Dr. Victor Babes” of Bucharest, 030303 Bucharest, Romania; dr_roxana_radoi@yahoo.com; 5Clinical Infectious Disease Hospital of Constanta, 900178 Constanta, Romania; 6Romanian Academy of Medical Sciences, 030167 Bucharest, Romania; 7Academy of Romanian Scientists, 050045 Bucharest, Romania

**Keywords:** atherosclerosis, acute coronary syndrome, HIV, chronic inflammation, CD4+ lymphocytes

## Abstract

People living with HIV infection are at high risk for cardiovascular events due to inflammation and atherosclerosis. Also, some antiretroviral therapies may contribute to the risk of cardiovascular complications. Immune status is highly dependent on the level of lymphocyte T helper CD4+. There are data suggesting that immune status and CD4+ cell count may be involved in the development of cardiovascular complications in these patients. Our study is longitudinal and retrospective and included a total number of 50 patients with HIV infection associated with acute coronary syndrome, divided into two subgroups based on the nadir of CD4+ cells. This study analyzes the relationship between the immune status of HIV patients, assessed by the nadir of the CD4+ T-cell count, and the outcome of these patients. Also, secondary endpoints were the assessment of the magnitude of coronary lesions and of thrombotic and bleeding risk assessed by specific scores. Clinical and biological parameters and also the extension and complexity of coronary lesions were assessed. Although patients with poor immune status had more complex coronary lesions and increased operative risk and bleeding risk at one year, this was not associated with significant differences in major adverse cardiac and cerebrovascular events at the 30-day and 1-year outcomes.

## 1. Introduction

Highly active antiretroviral therapy (HAART) modified the course of HIV infection, turning it into a manageable chronic disease, with people living with HIV (PLWHs) having a longer survival and an improved quality of life. Before the antiretroviral era, the most frequent presentations of HIV-associated cardiovascular disease were pericardial disease, dilated cardiomyopathy, endocarditis or pulmonary hypertension. At present, many patients with HIV infection develop cardiovascular complications related to atherosclerosis, and many studies emphasize that people living with HIV have an increased risk of atherosclerosis [1,2].

Complex pathophysiological mechanisms contribute to the development and accelerated progression of atherosclerosis in patients diagnosed with HIV infection. There is an increased cardiovascular risk profile in HIV patients due to a higher prevalence of traditional risk factors (dyslipidemia, smoking, diabetes mellitus, hypertension, and drug abuse) and also to specific risk factors (inflammation, endothelial dysfunction, coagulation abnormalities, and antiretroviral therapy) [3,4,5]. Coronary artery disease is one of the most important atherosclerotic complications for patients with HIV infection [3], with data from the literature indicating a 1.5-fold to 2-fold increased risk of developing acute myocardial infarction [5,6] compared to the general population. Clinical and paraclinical characteristics of acute coronary syndrome in people living with HIV, as well as the outcome factors, are different in various studies [1,7,8,9,10,11,12,13,14].

There are studies that outlined that the risk of myocardial infarction in these patients remains significantly high even after the increase in CD4+ levels and the decrease in the viral burden [15]. The initiation of HAART in PLWHs must be performed early, regardless of the CD4 values or viremia, and there are studies that reported reductions of over 40% in the risk of death or an acquired immune deficiency syndrome-defining event [16,17,18]. The SMART trial (Strategies for Management of Antiretroviral Therapy) compared CD4-count-guided treatment interruptions with continuous ART and demonstrated a 70% increase in cardiovascular adverse events in patients who discontinued targeted treatment [18,19].

The present study aims to point out the implication of the immunological condition, assessed by the CD4+ nadir, on the outcome of HIV-infected patients with the associated acute coronary syndrome, expressed by the complexity and severity of the coronary lesions at angiography.

## 2. Materials and Methods

### 2.1. Study Design

We investigated the relationship between a poor immune status, expressed by CD4 T cell count and the risk of cardiovascular adverse events and prognosis in HIV patients who developed acute coronary syndrome. Also, we hypothesized that there may be a correlation between the CD4 nadir and the severity of coronary lesions at angiography. We conducted an observational longitudinal, retrospective, case-controlled, and multicentric study using medical databases from the main departments of cardiology and infectious disease from Bucharest and Constanta, Romania, over a period of 13 years between October 2009 and October 2022.

The entire study group of patients diagnosed with HIV infection and acute coronary syndrome consisted of 50 patients who were divided into two subgroups based on the nadir of CD4+ lymphocyte count. The study group consisted of HIV patients diagnosed with acute coronary syndrome, age > 18 years old, with CD4+ count available. Exclusion criteria were patient refusal and missing CD4+ count data. Four patients were excluded because of missing data regarding CD4+ cell count; further analysis was performed on the remaining 46 patients. Subgroup A consisted of 27 patients with CD4+ nadir ≤ 200 cells/mm^3^, and subgroup B included 19 patients with CD4+ nadir > 200 cells/mm^3^. The diagnosis of acute coronary syndrome was made based on clinical evaluation, the presence of ischemia or necrosis on ECG, and the specific dynamic of myocardial necrotic enzymes. On this basis, all three major types of acute coronary syndrome were represented in the study group (unstable angina and ST-segment or non-ST segment elevation myocardial infarction).

Our study had a composite primary endpoint that consisted of in-hospital mortality and the analysis of 30-day and 360-day major adverse cardiac and cerebrovascular events (MACCE) consisting of recurrent acute coronary syndrome, heart failure requiring hospitalization, cardiac death, and stroke related to the CD4+ nadir. The secondary endpoints were the extension and complexity of coronary lesions in relation to the immune status and also the analysis of the risk scores for thrombotic and bleeding events in acute coronary syndrome for the entire study group.

For all patients included in the study, we analyzed clinical, biological, echocardiographic, and angiographic data. We assessed the presence of traditional cardiovascular risk factors (smoking, hypertension, diabetes, obesity, and history of coronary artery disease, including the type and severity of the acute coronary syndrome at presentation) and biological parameters (serum hemoglobin, creatinine CKD-EPI Equations for Glomerular Filtration Rate, total cholesterol, triglycerides, natremia, kalemia, and myocardial enzymes CK and CK-MB). Data regarding morphological and functional parameters were also analyzed with echocardiography: the presence of significant valvulopathies and pericardial effusion and left ventricle aneurysm, left ventricular ejection fraction, and the probability criteria for pulmonary hypertension.

We used several scores to evaluate the clinical prognosis, the complexity of coronary involvement, the decision making for the method of revascularization, in-hospital mortality after cardiac surgery, and the risk for bleeding complications in HIV patients on dual antiplatelet therapy.

The risk for in-hospital mortality was assessed only in HIV patients with non-ST elevation acute coronary syndrome using the GRACE score (Global Registry of Acute Coronary Events), which is based on clinical and biological parameters: age, heart rate, systolic blood pressure, cardiac arrest, Killip class, segment deviation, serum creatinine, and cardiac biomarkers. There are three categories of GRACE risk score: low risk (109 points), intermediate risk (109–140 points), and high risk (>140 points).

To evaluate the extent and severity of coronary lesions at coronary angiography, we used SYNTAX I and SYNTAX II (SYNergy between percutaneous coronary intervention with TAXus and cardiac surgery) scores. The SYNTAX I score predicts the risk associated with percutaneous intervention (PCI) or surgical revascularization (CABG). It represents the sum of the points assigned to each lesion found in the coronary arteries with >50% diameter narrowing in vessels over 1.5 mm diameter. A SYNTAX I score ≥ 23 suggests a higher risk for PCI compared to surgical revascularization [20]. SYNTAX II score indicates 4-year mortality risk for the two types of revascularization (PCI or CABG).

Euro-SCORE II (European System for Cardiac Operation Risk Evaluation) is the most utilized risk stratification tool to predict in-hospital mortality after cardiac surgery. There are three categories of risk in the classification of Euro-SCORE II: low risk (0–2%), moderate risk (2–5%), and high risk (>5%).

The risk for bleeding complications was assessed with the PRECISE-DAPT score (PREdicting bleeding Complications In patients undergoing Stent implantation and subsEquent Dual Anti Platelet Therapy) and derived TIMI scores for major and minor bleeding complications at 12 months. The PRECISE-DAPT score is applicable at patient discharge and estimates the risk of bleeding at 1 year from the event of acute coronary syndrome. For patients with high bleeding risk (PRECISE-DAPT score ≥ 25), long-term administration of dual antiplatelet therapy was associated with major bleeding events and no ischaemic benefit. Conversely, for patients with low bleeding risk (PRECISE-DAPT score < 25), long-term administration was associated with a significant reduction in thrombotic events (stent thrombosis and target vessel revascularization), with no increase in bleeding [21].

### 2.2. Statistical Analysis

The database was created in Microsoft Office Professional Plus 2016 (Microsoft Excel) and then exported to Statistical Package for Social Sciences (SPSS) to be processed. The chi^2^ test and Student’s *t*-test were used to evaluate statistical significance between categorical and continuous variables.

The uniformity of both subgroups (A and B) was evaluated for basic characteristics at presentation, and analysis was performed according to the characteristics of the variables followed. Data analysis was transposed as a table, presenting absolute values for presence and “*p*” value for categorical variables. All the differences were considered significant at a two-tailed *p*-value < 0.05. We used logistic regression to analyze the relationship between various independent predictors and major adverse cardiac and cerebrovascular events.

We used a multivariate logistic regression model to analyze the association between various independent variables and the MACCE binary endpoint. The endpoint was adjusted for smoking, diabetes mellitus, culprit lesion, type of ACS, and macrocytosis in its unit of measure. Odds ratios and corresponding 95% confidence intervals were calculated. The 95% confidence intervals were not adjusted for multiple testing and should not be used to infer definitive effects.

## 3. Results

### 3.1. Clinical and Biological Features

Demographic data were similar between the two subgroups; the mean age was 50 years (SD 12.67 years) in subgroup A versus 49.32 years (SD 10.82 years) in subgroup B, with males being equally represented (Table 1). The analysis of cardiovascular risk factors found no significant differences between the two subgroups. Even without statistical significance, we observed that only hypertension was more prevalent in the subgroup of patients with a compromised immune status (subgroup A). In our study, atypical angina at presentation was more prevalent in patients with a CD4+ nadir ≤ 200 cells/mm^3^. Although peripheral artery disease and chronic kidney disease were obviously more prevalent in absolute values in subgroup A (twice as much as in subgroup B), the data did not reach statistical significance (Table 1).

The mean time interval between CD4+ nadir determination and the onset of acute coronary syndrome was similar between the two subgroups (14.07 in subgroup A vs. 13.05 years in subgroup B) (Figure 1). The use of different classes of antiretroviral therapy was not significantly different between the two subgroups (*p* > 0.05) (Figure 2). The mean time interval between therapy exposure and the onset of acute coronary syndrome had no differences (13.85 years in subgroup A versus 11.15 years in subgroup B) (Figure 3).

For the entire study group, the following biomarkers were analyzed: HIV viral load, serum hemoglobin, mean corpuscular volume, glomerular filtration rate (CKD-EPI), blood levels of sodium and potassium, myocardial necrosis markers (creatin kinase and creatin kinase-MB). Comparative analysis between the two subgroups showed no statistical differences, except for hyperkalemia, which was significantly prevalent in subgroup A (*p* = 0.021). (Table 1).

The analysis of the type of acute coronary syndrome at presentation found no differences in the two subgroups. For patients with acute myocardial infarction with or without ST-segment elevation, the Killip class was also no different (Table 2).

### 3.2. Angiographic Features

Culprit lesions and associated coronary lesions at angiography showed no significant differences. Analysis of the extension of coronary artery disease revealed that the number of coronary lesions was similar in the two subgroups. Nevertheless, the only significant difference found in coronary angiography was related to the complexity of the coronary artery disease assessed by the SYNTAX I score. In subgroup A, 40.9% of patients had SYNTAX I score ≥ 23 points, and in subgroup B, all the patients had SYNTAX I score < 23 points (*p* = 0.013). The indication for coronary artery bypass graft (CABG) and its achievement was similar in both subgroups, as was the rate of intrastent restenosis or the presence of venous graft stenosis (Table 3).

### 3.3. Electrocardiographic and Echocardiographic Characteristics

The analysis of the presence of electrocardiographic changes indicated the presence and the pattern of myocardial ischemia and supraventricular and/or ventricular arrhythmia. During echocardiography, parameters of cardiac function and morphology were determined, such as left ventricular systolic function (left ventricular ejection fraction), left ventricular diastolic function, the presence and severity of functional mitral regurgitation (the presence of moderate/severe mitral regurgitation was considered relevant), the presence of a left ventricular aneurysm, pericardial effusion, and the probability of pulmonary hypertension (Table 4).

Electrocardiographic changes were quite similar between the two study subgroups, although, in subgroup A, supraventricular and ventricular arrhythmias were more frequently present. Also, during echocardiography, we found no significant differences, although severe left ventricular systolic dysfunction and functional mitral regurgitation (probably due to myocardial ischaemic changes and/or left ventricular enlargement) were more frequently seen in patients in subgroup A compared to those in subgroup B, in whom diastolic dysfunction was more common. Pulmonary hypertension (patients with intermediate or high probability) was also more present in patients in subgroup A, but without statistical significance. Left ventricular remodeling and pericardial effusion were similar in both subgroups (Table 4).

### 3.4. The Risk of Thrombotic and Bleeding Events

In general, antithrombotic treatment is mandatory for the first year after acute coronary syndrome is treated invasively or conservatively, but there are situations in which dual antiplatelet therapy needs to be extended. Therefore, the balance between thrombotic and bleeding risk should be carefully assessed in all patients. In our study group, an elevated DAPT (Dual Anti Platelet Therapy) score of over 2 points was present in both subgroups of patients, with no significant differences, but patients in subgroup A had an obvious tendency for increased estimated risk of bleeding complications, with greater PRECISE-DAPT score compared with patients in subgroup B (30% vs. 5.88%, *p* = 0.061) (Table 5).

### 3.5. Outcome and Prognosis

Even if in-hospital mortality was almost three times greater in subgroup A, it did not reach statistical significance, most probably due to the small number of patients in both subgroups. Major adverse cardiac and cerebrovascular events (MACCE) at 30 days and 360 days (acute coronary syndrome recurrence, heart failure requiring hospitalization, cardiovascular death, and stroke) and cumulative MACCE did not differ between the two subgroups (Table 6).

The analysis of continuous variables in the two subgroups (Table 7) showed significant differences in the mean level of CD4+ count at the onset of acute coronary syndrome, with patients in the subgroup with a CD4+ nadir ≤ 200 cells/mm^3^ having significantly lower mean values (476.74 [SD 242.53] vs. 805.74 [SD 384.34]; *p* = 0.001). For other biological markers, such as HIV viral load, serum hemoglobin, mean corpuscular volume, serum electrolytes, lipid profile, or myocardial necrosis markers, no significant differences were found.

In our study, the SYNTAX I score was different, with statistically significantly higher values for patients in subgroup A (*p* = 0.022). The SYNTAX II score was also significantly different between the two subgroups, with patients in subgroup A also having significantly higher values (*p* = 0.046). As concerns the risk of percutaneous coronary intervention 4 year-mortality, derived from the SYNTAX II score, we also noticed significant differences, with higher mean rates for patients in subgroup A (*p* = 0.036) (Table 7).

The risk of bleeding complications, assessed by the PRECISE-DAPT risk score, was significantly higher for patients with low levels of CD4+ cells (*p* = 0.037), and TIMI risk for major and minor bleeding complications at 12 months was also significantly higher in subgroup A (*p* = 0.016) (Table 7).

### 3.6. Multivariable Logistic Regression Analysis

A multivariable logistic regression analysis was performed to ascertain the effects of different independent variables. The analysis included all the patients in the study group except the patients in whom the nadir value of CD4+ cells was lacking. No significant cardiac adverse events were noticed at 30 days of analysis (Table 8) (Figure 4).

In the multivariate regression analysis for 360-day MACCE, we did not find significant changes (Table 9). Nevertheless, we noticed that patients with diabetes, OR 2.070 (95% CI 0.888–70.661, *p* = 0.064) and macrocytosis, OR 1.309 (0.787–17.407, *p* = 0.098), have an increased tendency to develop major adverse cardiac and cerebrovascular events at 360 days (Figure 5).

## 4. Discussion

HIV infection and atherosclerosis each represent important health concerns. The presence of atherosclerotic complications in HIV patients influences their prognosis. Mechanisms of HIV-related atherosclerosis are not very different, but HIV infection itself and the association of some antiretroviral therapies may contribute to premature development and acceleration of atherosclerotic changes. Similar to antineoplastic therapies for patients with different types of cancer, antiretroviral therapies represent huge progress in modern medicine, increasing survival for both categories of patients. Nevertheless, both therapies may contribute to the development of important cardiovascular complications that have a great influence on their prognosis.

CD4 T cell count is influenced by viral and various individual factors. The decline of CD4 count is associated with the progression of the disease and poor prognosis in correlation with the viral load. Concerning the relationship between CD4 count and the risk of cardiovascular complications, there are no large randomized studies to date. The 2021 ESC Guidelines on cardiovascular disease prevention highlight that patients with HIV infection develop more frequently atherosclerotic peripheral artery disease and coronary artery complications [22] compared to the general population and that the risk is two-fold greater in those patients with CD4+ < 200 cells/mm^3^. On the other hand, in people living with HIV with sustained levels of CD4+ > 500 cells/mm^3^, the relative risk is similar to the risk of HIV-negative patients [23]. One study found that even a small decrease in the level of CD4+, beneath 500 cells/mm^3^, is independently associated with the risk of cardiovascular diseases [24].

Low levels of CD4+ cells are strongly related to cardiovascular complications that are heterogeneous. The overall impact of the immune status on the evolution and prognosis of acute coronary syndrome in PLWHs is still unclear. Uncontrolled HIV infection (characterized by persistently low levels of CD4+ and high levels of viral burden) has a direct influence on the metabolism of HDL-cholesterol and accelerated atherothrombosis. This mechanism explains, at least in part, higher rates of recurrent ischemic events in these patients [7,25,26]. The lowest level of CD4+ (the nadir of CD4+) is one of the most important factors strongly correlated with the presence of less calcified, highly unstable atherosclerotic plaques [27,28]. However, severe advanced atherosclerosis, such as porcelain aorta, has been reported in young persons infected with HIV since childhood [29].

Clinical and experimental studies indicate that increased oxidative stress in HIV infection, sometimes exacerbated by some antiretroviral drugs, is associated with a reduction in CD4+ T-cell level, cytotoxicity, and endothelial dysfunction, contributing to atherosclerotic changes [30]. Other mechanisms, like activation of endoplasmic reticulum stress, may lead to macrophage apoptosis and apoptosis of smooth muscle cells, contributing to the decrease in collagen synthesis in the fibrous cap, which eventually leads to plaque instability and disruption [30]. Abnormal immune system response promotes inflammation and inflammatory cytokines (TNFα, IL-6, and IL-1β) that also contribute to the initiation and progression of atherosclerosis and also to the depletion of T cells. Also, HIV infection suppresses autophagy, which is an important protective factor against oxidative stress and an inhibitor of apoptosis. The individual contribution of these mechanisms to atherosclerosis in this particular setting of HIV infection, the influence of antiretroviral therapies on these mechanisms, and also the influence of comorbidities remain to be established by further analysis [30].

In the present case–control study, we included patients with HIV infection who developed acute coronary syndrome, regardless of the type of presentation. At this moment, we are not aware of studies that strictly compared patients with HIV infection and the associated acute coronary syndrome based on the CD4+ nadir cell count in the literature.

The immune status and HIV viral load can predict risk in PLWHs, and the following correlations were reported in the literature [31]: a. recent low CD4+ values with cardiovascular risk and prognosis of patients who have developed cardiovascular diseases [15,24,32]; b. a CD4+ nadir < 350 cells/mm^3^ with endothelial dysfunction [33]; c. a low CD4+ level with soft, non-calcified atherosclerotic plaques, with risk of instability [34]; and d. delayed immune recovery after initiating antiretroviral therapy with increased rates of cardiovascular events [35].

Comparing the two groups based on the prevalence of cardiovascular risk factor (CVRF), no statistically significant difference was identified for traditional contributors, including diabetes and the history of coronary disease, which are equally important predictors for severe myocardial complications. A study that analyzed certain CVRF for patients with HIV infection with a CD4+ nadir < 350 cells/mm^3^ pointed out a statistically significant higher prevalence of dyslipidemia, but a non-significant higher prevalence of hypertension and diabetes mellitus compared to HIV patients with a nadir of CD4+ ≥ 350 cells/mm^3^ [33]. The comparison with our study may not be relevant due to the fact that the mentioned study did not include HIV patients with the associated acute coronary syndrome, and the reference value for the nadir of CD4+ in this study was higher (350 cells/mm^3^).

Acute coronary syndrome with atypical presentation is common in PLWHs [14,36,37], although there are no available exact data on the prevalence of atypical angina in these patients, especially related to the CD4+ cell count. In the present study, a significant difference was found in the percentage of patients who had atypical forms of angina at presentation, although not significant. Peripheral atherosclerotic determinations were twice as prevalent in patients with a low CD4+ nadir, indicating more severe atherosclerotic changes in patients with severely compromised immune status. When compared to the literature, published studies showed that PLWHs have a 19% higher risk of developing peripheral artery disease compared to the general population, and the risk doubles in HIV patients with CD4+ < 200 cells/mm^3^ compared to those with CD4+ ≥ 500 cells/mm^3^ [23,38].

The biological profile of patients in the two groups was similar. Among the current biomarkers analyzed, we found a statistically significant difference in the prevalence of hyperkalemia and impaired kidney function. Data from the literature indicate that among the major causes of hyperkalemia in HIV patients are adrenal insufficiency, hyporeninemic hypoaldosteronism, concomitant administration of trimethoprim or pentamidine, or impaired renal function [39,40]. Only a relatively small number of patients showed that CD4+ cell count is not a predictor for the presence or absence of adrenal insufficiency [41]. Regarding renal function impairment, lower glomerular filtration rate was more frequently present in subgroup A, even if it was only approaching the statistical significance in the analysis of discrete variables. Not surprisingly, patients with a nadir of CD4+ < 200 cells/mm^3^ had significantly lower CD4+ cell count at the diagnosis of acute coronary syndrome.

Contrary to what we expected, the analysis of coronary involvement (coronary culprit lesion and associated non-culprit lesions) in the two groups did not show significant statistical differences. It is noteworthy that the involvement of the left main coronary artery was more frequent in patients with a low CD4+ nadir. Our results are in accordance with other studies from the literature that also found equal prevalence between PLWHs and non-HIV patients [1,42], with no reported link between immune status and left main coronary artery impairment. Also, there are some clinical cases reported in which patients with left main coronary artery disease had a reasonable level of CD4+ cells, but did not mention the CD4+ nadir [36,37]. Although there is a significant percentage difference in the indication of surgical revascularization, which is more than twice as high for patients in subgroup A, as well as its implementation, statistical significance was not reached. The fact that patients in subgroup A had statistically significantly higher SYNTAX I scores than patients in subgroup B shows an increased extension of coronary lesions among them. In addition, the significantly higher SYNTAX II scores in these patients, a score that includes other paraclinical data such as creatinine clearance, left ventricular ejection fraction, and the presence of lung disease and peripheral artery disease, shows an increased risk in these patients and a tendency for surgical myocardial revascularization indication instead of percutaneous angioplasty. PLWHs with a CD4+ nadir ≤ 200 cells/mm^3^ have a higher operative risk than those with a CD4+ nadir > 200 cells/mm^3^, with a Euro-SCORE II score over three times higher.

The values of the GRACE score, designed to estimate the risk of in-hospital mortality following acute coronary syndrome without persistent ST-segment elevation, were not statistically different between the two groups. Similarly, the values of the TIMI score, which predicts the risk of death at 30 days following acute coronary syndrome with persistent ST-segment elevation, were approximately equal in the two groups. The PRECISE-DAPT score that predicts bleeding complications in HIV patients requiring dual antiplatelet therapy after revascularization by percutaneous coronary intervention and the risk scores derived from it, like TIMI score, were significantly higher for patients with a low CD+ nadir in our study. At this moment, we found no data to compare our findings regarding the relationship between the risk of bleeding and the CD4+ nadir with other studies.

It is known that the magnitude of the impaired immune status of HIV patients is related to the risk of developing adverse cardiovascular events [24,43,44]. Both the nadir of CD4+ and CD4+ counts represent independent cardiovascular risk factors [44]. The data published underline that patients with CD4+ count lower than 350 cells/mm^3^ have a higher incidence of cardiovascular events (HR 1.58; 95% CI 1.09–2.30) compared to those with CD4+ > 500 cells/mm^3^ (HR 2.64; 95% CI 1.66–4.20) [24], and the incidence is even higher in patients with CD4+ < 200 cells/mm^3^ (HR 0.67; 95% CI 0.22–2.08) compared to 86,321 non-HIV patients (*p* < 0.001) [43]. In our study, PLWHs with a nadir of CD4+ ≤ 200 cells/mm^3^ had an almost three times higher rate of in-hospital mortality than those with CD4+ > 200 cells/mm^3^; although this difference did not reach statistical significance (*p* = 0.305), this is most probably influenced by the relatively small absolute number of patients. Regarding the MACCE rate at 30 days after discharge, patients with a low CD4+ nadir had more frequent acute coronary syndrome recurrence, more frequent hospitalizations for heart failure, and more deaths from cardiovascular causes, with no relevant differences. Also, follow-up at 360 days indicates similar rates of acute coronary syndrome recurrence and cardiovascular deaths, and no significant differences were found between the two subgroups in MACCE at 360 days. Other, more recent studies showed that the only independent predictor for recurrence of coronary syndrome is HIV infection per se and that other analyzed parameters, such as the nadir of CD4+, viral load, CD4/CD8 radio, or duration of HIV infection, had no significant impact [10,43]. Among the adverse predictors analyzed in the logistic regression for MACCE at 30 and 360 days, none of smoking, diabetes mellitus, culprit lesion, type of ACS, or macrocytosis had a statistically significant effect. Nevertheless, it is to be mentioned that for MACCE at 360 days, diabetes mellitus and macrocytosis nearly approached statistical significance.

The impairment of the immune status in PLWHs, quantified by the CD4+ nadir, may have an important role in the occurrence of cardiovascular complications. However, low values of CD4+ count, in the presence of a cluster of cardiovascular risk factors, may also explain in part the occurrence of coronary artery disease in these patients, and it is most likely that other molecular mechanisms may also be involved and are subject to further analysis.

The Global Burden of Cardiovascular Diseases Collaboration is an ongoing effort to increase the quality and availability of public health [45]. More centralized data on ischemic heart disease in PLWHs and poor immune status would be very useful.

### 4.1. Study Limitations

We are aware of some limitations that could have influenced our results. The study is retrospective and involved a relatively small number of patients with HIV infection and the associated acute coronary syndrome. Clinical progression could not be recorded for some patients due to a lack of contact data (especially for those who were hospitalized a long time ago), leading to some patients being lost to follow-up. A complete lipid profile could have been useful to better characterize the cardiovascular risk. We are also aware that CD4+ count describes only in part the immune status of these patients, being at the same time an important parameter used for the Centers for Disease Control (CDC) classification system for HIV infection.

### 4.2. Future Perspectives

Patients with HIV infection have higher cardiovascular risk compared to the general population. Immune status, as indicated by a low CD4+ cell count, is closely related to cardiovascular impairment in general. Recently developed practice guidelines [22] report a two-fold increase in cardiovascular risk for patients with CD4+ values ≤ 200 cells/mm^3^, a risk that decreases as immune status improves, with patients with sustained values of ≥500 cells/mm^3^ having a relatively similar risk to HIV-negative patients. PLWHs have diffuse and accelerated atherosclerotic disease with multifactorial etiology and mechanisms that are incompletely understood. Both the severity of coronary lesions and the aspect of the atheroma plaque have a significant influence on these patients. Intracoronary imaging, such as intravascular ultrasound (IVUS), is highly useful in characterizing coronary atherosclerotic disease and establishing therapeutic management. In a study that compared the ultrasonographic characteristics of coronary atheromatous plaques in 66 HIV patients with acute coronary syndrome with those of 120 non-HIV patients, patients reported higher rates of soft, non-calcified plaques with features of instability in HIV patients, which were associated with higher rates of adverse cardiovascular events and recurrent ACS, especially for patients with CD4+ values ≤ 200 cells/mm^3^ [27]. By completing intracoronary investigations with optical coherence tomography (OCT), useful data can be obtained by analyzing the physiopathology of intracoronary atherosclerosis in native vessels. For patients with recurrent ACS due to intrastent restenosis, either by thrombosis, neointimal hyperplasia or neo-atherosclerosis, OCT represents the gold standard for determining the mechanism. Quantifying coronary stenosis at the hemodynamically significant limit subjectively assessed by classic angiography, as well as new intracoronary hemodynamic evaluation methods such as resting full-cycle ratio (RFR), fractional flow reserve (FFR), or instantaneous wave-free ratio (iFR), would clarify the notion of diffuse vascular involvement, highlighting coronary microvascular involvement, which is frequently encountered in these patients. Analyzing data obtained through intracoronary imaging, determining the number of CD4+ cells at the time of ACS, and quantifying the potential atherogenicity of antiretroviral medication could lead to beneficial correlations and establish a specific approach.

Since the number of patients who have both acute coronary syndrome and human immunodeficiency virus infection is relatively small, multicenter randomized trials that include a significant number of patients analyzed in tertiary cardiology centers capable of extensively investigating coronary circulation will be particularly useful.

## 5. Conclusions

In HIV patients with an affected immune status, which in the present study was expressed by lower values of the CD4+ nadir, the extension of coronary artery disease assessed by SYNTAX scores was significantly higher; these patients also have significantly lower mean values of CD4+ counts at similar cardiovascular risk factors, coronary lesions, and baseline characteristics.

In these patients, we also observed a higher risk of major and minor bleeding at one year. Nevertheless, the CD4+ nadir did not influence MACCE at 30 days and 1 year in our study.

Based on the results presented, we consider that for patients with HIV infection, particularly those with poor immune status, the risk for cardiovascular complications should be carefully monitored. Therefore, evaluation of the presence of cardiovascular risk factors, lifestyle measures, and treatment initiation, if indicated, should be performed as early as possible. Strict and periodic clinical, biological, and non-invasive cardiovascular imaging surveillance of patients with chronic HIV infection for signs and symptoms of coronary heart disease should be recommended. The constant improvement in the immunological condition may also lower the risk for cardiovascular events in these patients. Also, it is important to carefully assess potential treatment interactions in these patients.

## Figures and Tables

**Figure 1 diagnostics-13-02682-f001:**
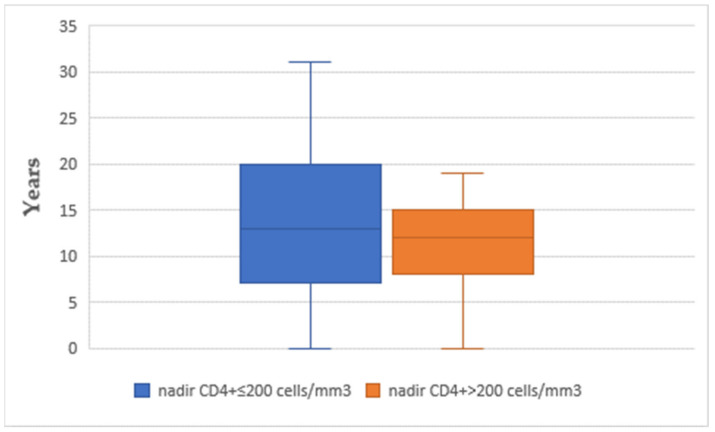
The time interval between the CD4+ nadir and the development of acute coronary syndrome (ACS).

**Figure 2 diagnostics-13-02682-f002:**
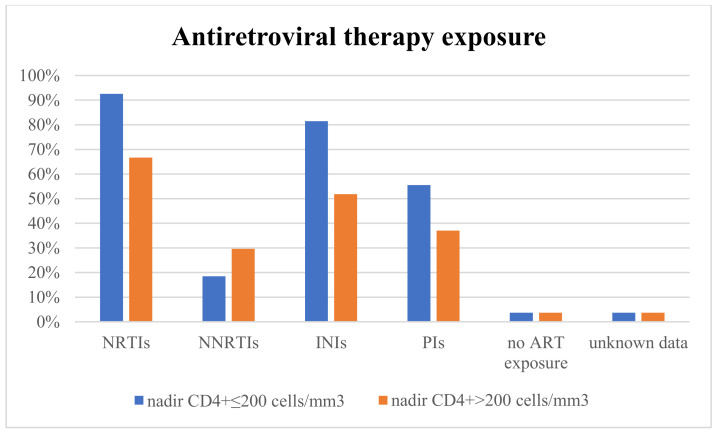
Antiretroviral therapy exposure. NRTIs—nucleoside reverse transcriptase inhibitors; NNRTIs—non-nucleoside reverse transcriptase inhibitors; INIs—integrase inhibitors; and PIs—protease inhibitors.

**Figure 3 diagnostics-13-02682-f003:**
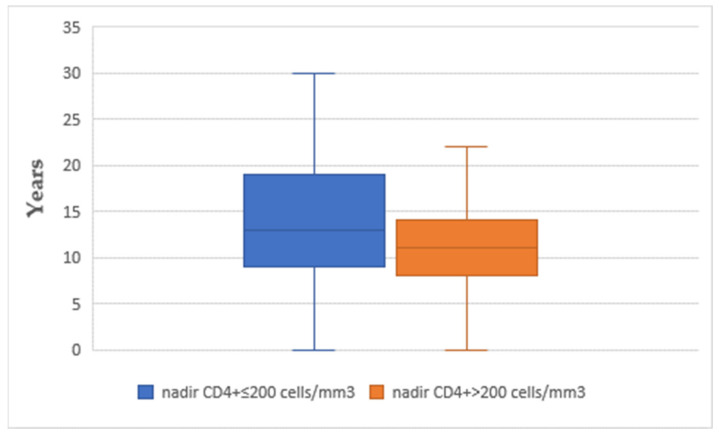
The time interval between the antiretroviral therapy (ART) and the occurrence of acute coronary syndrome.

**Figure 4 diagnostics-13-02682-f004:**
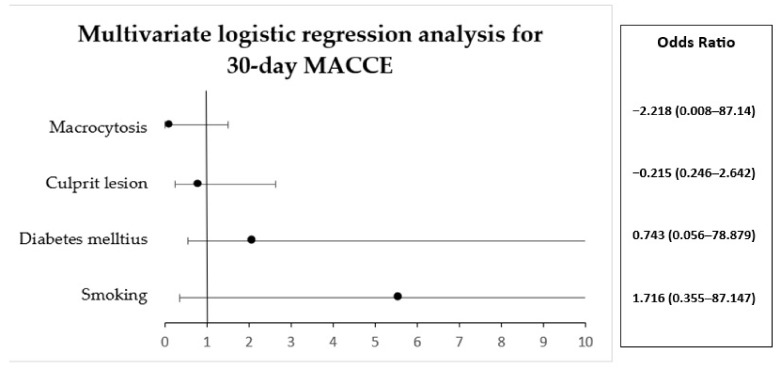
Multivariate logistic regression analysis for 30-day MACCE.

**Figure 5 diagnostics-13-02682-f005:**
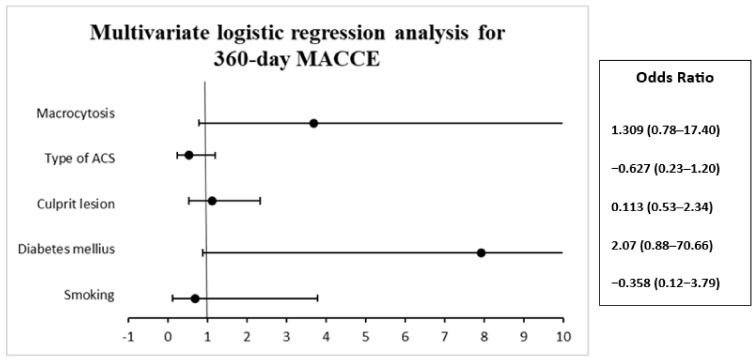
Multivariate logistic regression analysis for 360-day MACCE.

**Table 1 diagnostics-13-02682-t001:** The analysis of statistical basic descriptive differences for discrete variables in the two subgroups based on cardiovascular risk factors, clinical history, and clinical and biological parameters.

	CD4+ Nadir ≤ 200 Cells/mm^3^	CD4+ Nadir > 200 Cells/mm^3^	*p* Value
Males, *n*%	25 (92.6%)	18 (94.7%)	0.774
Detectable HIV viral load, *n*%	4 (13.79%)	6 (35.29%)	0.179
Smoking, *n*%	16 (61.5%)	12 (63.2%)	0.911
Hypertension, *n*%	17 (65.4%)	11 (57.9%)	0.608
Dyslipidemia, *n*%	20 (76.9%)	15 (78.9%)	0.871
Treated dyslipidemia, *n*%	12 (60%)	10 (66.6%)	0.847
Diabetes mellitus, *n*%	4 (15.4%)	5 (26.3%)	0.365
Obesity, *n*%	1 (3.8%)	3 (15.8%)	0.164
CAD history, *n*%	8 (29.6%)	6 (31.6%)	0.887
Atypical angina, *n*%	7 (28%)	3 (16.7%)	0.480
Peripheral artery disease, *n*%	9 (36%)	3 (16%)	0.191
Chronic kidney disease, *n*%	6 (23.1%)	2 (10.5%)	0.435
Anemia, *n*%	9 (33.3%)	4 (21.1%)	0.510
Macrocytosis, *n*%	9 (33.3%)	10 (52.6%)	0.233
Creatinine clearance < 60 mL/min/m^2^ (CKD-EPI), *n*%	6 (22.2%)	1 (5.6%)	0.215
Hyponatremia, *n*%	4 (14.8%)	3 (15.8%)	0.927
Hypernatremia, *n*%	1 (3.7%)	0	1
Hypokalemia, *n*%	3 (11.1%)	3 (15.8%)	0.642
Hyperkalemia, *n*%	4 (22.2%)	0 (0%)	0.021
Elevated myocardial necrosis markers, *n*%	24 (88.9%)	15 (78.9%)	0.424

CAD—coronary artery disease.

**Table 2 diagnostics-13-02682-t002:** The analysis of statistical basic descriptive differences for discrete variables in the two subgroups in relation to clinical features of acute coronary syndrome.

	CD4+ Nadir ≤ 200 Cells/mm^3^	CD4+ Nadir > 200 Cells/mm^3^	*p* Value
STEMI, *n*%	11 (40.7%)	8 (42.1%)	0.914
NSTEMI, *n*%	6 (22.3%)	5 (26.3%)	
Unstable angina, *n*%	10 (37%)	6 (31.6%)	
Killip I class, *n*%	13 (76.5%)	12 (92.3%)	0.464
Killip II class, *n*%	0 (0%)	0 (0%)	
Killip III class, *n*%	1 (5.9%)	0 (0%)	
Killip IV class, *n*%	3 (17.6%)	1 (7.7%)	

STEMI—ST-segment elevation myocardial infarction; NSTEMI—non-ST-segment elevation myocardial infarction.

**Table 3 diagnostics-13-02682-t003:** The analysis of statistical basic descriptive differences for discrete variables in the two subgroups in relation to acute coronary syndrome and coronary lesions.

	CD4+ Nadir ≤ 200 Cells/mm^3^	CD4+ Nadir > 200 Cells/mm^3^	*p* Value
Culprit lesion LAD, *n*%	10 (43.5%)	9 (56.2%)	0.504
Culprit lesion LCx, *n*%	3 (13%)	4 (25%)	
Culprit lesion RCA, *n*%	5 (21.7%)	2 (12.5%)	
Culprit lesion LM, *n*%	2 (8.7%)	0 (0%)	
Non-culprit lesion present, *n*%	3 (13%)	1 (6.2%)	
Associated lesion LAD, *n*%	9 (37.5%)	3 (20%)	0.215
Associated lesion LCx, *n*%	6 (25%)	4 (26.7%)	1
Associated lesion RCA, *n*%	9 (37.5%)	7 (49.7%)	0.740
Associated lesion LM, *n*%	4 (16.7%)	1 (6.7%)	0.631
Other associated lesions, *n*%	5 (20.8%)	4 (26.7%)	0.711
Single vessel disease, *n*%	10 (45.45%)	5 (33.33%)	0.704
Two-vessel disease, *n*%	5 (27.73%)	5 (33.33%)	
Three-vessel disease, *n*%	7 (31.18%)	5 (33.33%)	
SYNTAX I score ≥ 23 *p*, *n*%	9 (40.9%)	0 (0%)	0.013
Indication for coronary artery bypass graft surgery, *n*%	4 (16.7%)	1 (6.2%)	0.205
Performed coronary artery bypass graft surgery, *n*%	3 (12.5%)	0 (0%)	
Intrastent restenosis/venous graft stenosis, *n*%	4 (15.4%)	3 (16.7%)	1

LAD—left anterior descending artery; LCx—left circumflex artery; RCA—right coronary artery; LM—left main coronary artery; and SYNTAX I score—SYNergy between percutaneous coronary intervention with TAXus and coronary artery bypass surgery.

**Table 4 diagnostics-13-02682-t004:** The analysis of statistical basic descriptive differences for discrete variables in the two subgroups related to electrocardiographic and echocardiographic changes.

	CD4+ Nadir ≤ 200 Cells/mm^3^	CD4+ Nadir > 200 Cells/mm^3^	*p* Value
Ischaemic changes on ECG, *n*%	18 (69.2%)	15 (83.3%)	0.480
Supraventricular arrhythmia, *n*%	5 (19.2%)	2 (10.5%)	0.681
Ventricular arrhythmia, *n*%	5 (19.2%)	3 (15.8%)	1
LVEF ≤ 40%, *n*%	12 (44%)	7 (36.8%)	0.763
LVEF 41–49%, *n*%	5 (18.5%)	2 (10.5%)	0.682
LVEF ≥ 50%, *n*%	9 (34.6%)	8 (47.1%)	0.528
LV diastolic dysfunction, *n*%	11 (45.8%)	5 (29.4%)	0.344
Significant valvulopathy, *n*%	7 (29.2%)	3 (16.7%)	0.473
LV aneurysm, *n*%	2 (8.3%)	2 (11.1%)	1
Pericardial effusion, *n*%	1 (4.2%)	1 (5.6%)	1
Improbable PH, *n*%	20 (83.3%)	16 (94.1%)	0.525
Intermediate probability of PH, *n*%	3 (12.5%)	1 (5.9%)	
High probability of PH, *n*%	1 (4.2%)	0 (0%)	

ECG—electrocardiography; LVEF—left ventricular ejection fraction; LV—left ventricle; and PH—pulmonary hypertension.

**Table 5 diagnostics-13-02682-t005:** The analysis of statistical basic descriptive differences for discrete variables in the two subgroups related to the risk of thrombotic and bleeding events.

	CD4+ Nadir ≤ 200 Cells/mm^3^	CD4+ Nadir > 200 Cells/mm^3^	*p* Value
High DAPT score (≥2 points), *n*%	5 (29.41%)	3 (20%)	0.539
PRECISE-DAPT score ≥ 25 points, *n*%	6 (30%)	1 (5.88%)	0.061

**Table 6 diagnostics-13-02682-t006:** The analysis of statistical basic descriptive differences for discrete variables in the two subgroups related to outcome and prognosis.

	CD4+ Nadir ≤ 200 Cells/mm^3^	CD4+ Nadir > 200 Cells/mm^3^	*p* Value
In-hospital mortality, *n*%	4 (14.81%)	1 (5.26%)	0.305
Recurrent ACS at 30 days, *n*%	2 (8.69%)	1 (5.55%)	0.701
HF requiring hospitalization at 30 days, *n*%	2 (8.69%)	1 (5.55%)	0.701
Cardiovascular death at 30 days, *n*%	1 (4.34%)	0 (0%)	1
Stroke at 30 days, *n*%	1 (4.34%)	0 (0%)	1
Cumulative MACCE at 30 days, *n*%	6 (26.08%)	2 (11.11%)	0.229
Recurrent ACS at 360 days, *n*%	2 (11.8%)	2 (13.3%)	0.893
HF requiring hospitalization at 360 days, *n*%	0 (0%)	0 (0%)	-
Cardiovascular death at 360 days, *n*%	2 (11.8%)	2 (13.3%)	0.893
Stroke at 360 days, *n*%	0 (0%)	0 (0%)	-
Cumulative MACCE at 360 days, *n*%	4 (23.52%)	4 (26.66%)	0.837

ACS—acute coronary syndrome; HF—heart failure; and MACCE—major adverse cardiac and cerebrovascular events.

**Table 7 diagnostics-13-02682-t007:** Group statistics of basic descriptive data represented as continuous variables.

Variable	CD4+ Nadir (Cells/mm^3^)	Number	Mean Value	Standard Deviation	*p* Value
Age (years)	≤200	27	50.00	12.676	0.849
	>200	19	49.32	10.822	
CD4+ at the onset of the ACS (cells/mm^3^)	≤200	27	476.74	242.531	0.001
	>200	19	805.74	384.344	
Hemoglobin (g/dL)	≤200	27	13.9000	2.02123	0.254
	>200	18	14.5600	1.62463	
MCV (fL)	≤200	27	94.6481	11.25920	0.454
	>200	18	97.0150	8.63796	
MCHC (g/dL)	≤200	27	32.1185	3.47198	0.091
	>200	18	33.6256	1.51113	
Leucocytes (/mm^3^)	≤200	27	12,574.44	10,366.140	0.490
	>200	18	10,833.33	2497.156	
Thrombocytes (/mm^3^)	≤200	27	226,037.04	114,442.959	0.603
	>200	18	246,877.78	152,396.339	
Serum creatinine (mg/dL)	≤200	27	1.4207	1.22942	0.160
	>200	18	0.9767	0.56335	
Creatinine clearance (CKD-EPI) (mL/min/m^2^)	≤200	27	77.796	32.6624	0.059
	>200	18	97.083	32.5647	
Natremia (mmol/L)	≤200	25	139.996	3.4388	0.177
	>200	18	138.500	3.6340	
Kalemia (mmol/L)	≤200	25	4.1048	0.43413	0.153
	>200	18	4.3389	0.62269	
Total cholesterol (mg/dL)	≤200	26	181.35	47.935	0.413
	>200	18	168.67	53.005	
Triglycerides (mg/dL)	≤200	24	202.04	135.867	0.601
	>200	17	0.18	123.408	
Maximum value of CK (U/L)	≤200	21	702.71	1153.950	0.835
	>200	13	779.31	797.249	
Maximum value of CKMB (U/L)	≤200	26	118.81	126.024	0.361
	>200	16	86.44	76.772	
LVEF (%)	≤200	26	41.35	13.151	0.580
	>200	17	43.59	12.435	
SYNTAX I score (points)	≤200	22	17.068	13.3865	0.022
	>200	13	7.731	5.0275	
SYNTAX II score PCI (points)	≤200	22	31.432	20.6306	0.046
	>200	13	18.977	7.9748	
4-year mortality after PCI (%)	≤200	22	18.618	25.0906	0.036
	>200	13	3.254	2.0915	
SYNTAX II score after CABG (points)	≤200	22	19.818	17.2018	0.194
	>200	13	13.046	8.2357	
4-year mortality after CABG (%)	≤200	22	8.032	16.4931	0.194
	>200	13	2.015	1.3915	
EuroScore II (%)	≤200	22	3.8295454	4.905126062	0.056
	>200	13	1.1007692	0.7843836384	
GRACE score (points)	≤200	15	84.67	36.203	0.492
	>200	9	74.22	34.003	
GRACE death probability (admission–6 months) (%)	≤200	15	4.440	6.0655	0.445
	>200	9	2.756	2.8386	
TIMI risk score–mortality at 30 days after STEMI (%)	≤200	12	9.708	9.9378	0.519
	>200	8	7.063	6.6565	
PRECISE-DAPT score	≤200	22	18.00	13.956	0.037
	>200	15	9.73	5.548	
TIMI risk of major and minor bleeding at 12 months	≤200	22	1.6536	1.51422	0.016
	>200	15	0.6467	0.32264	
TIMI risk score of major bleeding at 12 months	≤200	22	0.8659	0.73168	0.013
	>200	15	0.3573	0.17119	

**Table 8 diagnostics-13-02682-t008:** Multivariate logistic regression analysis for MACCE at 30 days.

Parameter	B	S.E.	Wald	Df	Val. P	Exp(B)	95% C.I. for EXP(B)	
							Lower	Upper
Smoking	1.716	1.404	1.493	1	0.222	5.56	0.355	87.147
Diabetes mellitus	0.743	1.849	0.161	1	0.688	2.102	0.056	78.879
Culprit lesion	−0.215	0.605	0.126	1	0.723	0.807	0.246	2.642
Macrocytosis	−2.218	1.341	2.736	1	0.098	0.109	0.008	1.507

Independent variables: smoking, diabetes mellitus, culprit lesion, and macrocytosis.

**Table 9 diagnostics-13-02682-t009:** Multivariate logistic regression analysis for MACCE at 360 days.

Parameter	B	S.E.	Wald	Df	Val. P	Exp(B)	95% C.I. for EXP(B)	
							Lower	Upper
Smoking	−0.358	0.864	0.173	1	0.678	0.699	0.129	3.790
Diabetes mellitus	2.070	1.117	3.435	1	0.064	7.921	0.888	70.661
Culprit lesion	0.113	0.378	0.089	1	0.765	1.119	0.534	2.347
Type of ACS	−0.627	0.414	2.296	1	0.130	0.534	0.238	1.202
Macrocytosis	1.309	0.790	2.744	1	0.098	3.701	0.787	17.407

Independent variables: smoking, diabetes mellitus, culprit lesion, type of ACS, and macrocytosis.

## Data Availability

Data are unavailable.

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
