# Peer review of "Particular Aspects Related to CD4+ Level in a Group of HIV-Infected Patients and Associated Acute Coronary Syndrome"

_diagnostics, 2023, doi:10.3390/diagnostics13162682_

Round 1
Reviewer 1 Report
“Particular aspects related to CD4+ level in a group of HIV-infected patients and associated acute coronary syndrome”
The study includes important information on HIV infected patients who are at high risk for cardiovascular disease is an extensive and meticulous quality work. As for the experimental design and the innovative work I have no comments, these are only observations in terms of understanding the text and how to describe their results, I hope my comments serve to improve the reading of these and highlight them as they were without doubt of great relevance.
Abstract
ü I recommend that authors avoid extensive repetition, as authors mentioned two times (Our study is…, The aim of the present 21 study was…) and simply not write results in sentences but provide more condensed discussion with analysis data.
Introduction
ü Introduction does not set a problem and the need for this work properly. What has been done and what information is missing?
ü The authors have included unnecessary information which is not appropriate in this section. This information may be succinctly added to the discussion, when describing facts and comparing with the previous data.
ü Current scenarios for potential effects of the COVID-19 pandemic and the other potential impact of improvements preferentially targeting all ages survival due to this risk factor (HIV), could you please provide the some link between these.
Results and discussion
The data interpretation is well presented, the statistical significance of obtained data is little needed to improve. Please get some reference from Global Burden of Disease (GBD) studies.
The conclusion does not provide aspects mentioned in the title and body of the text as epidemiological perspectives, and clinical and diagnostic advances, and future plan according to socio-demographic predictor variables.
ü There are some typographical errors. These should be corrected.
Author Response
Dear reviewer,
We thank you for your observations that will make the study appropriate for publication. We took into consideration your observations and we did our best to make the changes required.
Some expressions and phrases were changed in order to avoid repetition.
In the Introduction section we precised more clear the problematic and the purpose of the study and some considerations were moved to the Discussion section.
Also, we included information according to Global Burden of Disease studies.
Our conclusions were adjusted to provide aspects mentioned in the title and body of the text.
Typographical errors were corrected.
Best regards.
Reviewer 2 Report
This manuscript by Bajdechi M investigates the implication of immune status assessed by CD4+ nadir on the outcome of HIV infected patients and associated acute coronary syndrome. For this study, they recruited 50 patients with HIV infection that associated an acute coronary syndrome, divided into two subgroups based on the nadir of CD4+ cells. They found that HIV patients with affected immune status with lower CD4+ nadir, the extension of coronary artery disease assessed by SYNTAX scores was significantly higher, these patients having also significantly lower mean values of CD4+ count at similar cardiovascular risk factors, coronary lesions and baseline characteristics. They also observed higher risk of major and minor bleeding at one year.
Overall, the analysis are conducted well and the manuscript is well written.
Author Response
Dear Reviewer,
We thank you for spending time reading and analyzing our study. We appreciate your observations and comments and improved some aspects in the text and we hope that the study is suitable for publication.
Best regards!
Round 2
Reviewer 1 Report
I appreciate the authors; they have improved the quality of the manuscript for wider dissemination of the findings and broader readership. Now I recommend this revised manuscript entitled “Particular aspects related to CD4+ level in a group of HIV-infected patients and associated acute coronary syndrome” for publication.